# Genetic Variants of Alcohol Metabolizing Enzymes and Alcohol-Related Liver Cirrhosis Risk

**DOI:** 10.3390/jpm11050409

**Published:** 2021-05-13

**Authors:** Pedro Ayuso, Elena García-Martín, José A. Cornejo-García, José A. G. Agúndez, José María Ladero

**Affiliations:** 1ARADyAL, Instituto de Salud Carlos III, University Institute of Molecular Pathology Biomarkers, UEx, 10003 Cáceres, Spain; elenag@unex.es (E.G.-M.); jagundez@unex.es (J.A.G.A.); 2ARADyAL, Instituto de Salud Carlos III Research Laboratory, IBIMA, Regional University Hospital of Málaga, UMA, 29010 Málaga, Spain; josea.cornejo@gmail.com; 3Service of Gastroenterology (Liver Unit), Hospital Clínico San Carlos, Universidad Complutense Medical School, 28040 Madrid, Spain; jladeroq@gmail.com

**Keywords:** alcohol-related liver disease, cirrhosis, single nucleotide variations, copy number variations, alcohol dehydrogenase

## Abstract

Alcohol-related liver disease (ARLD) is a major public health issue caused by excessive alcohol consumption. ARLD encompasses a wide range of chronic liver lesions, alcohol-related liver cirrhosis being the most severe and harmful state. Variations in the genes encoding the enzymes, which play an active role in ethanol metabolism, might influence alcohol exposure and hence be considered as risk factors of developing cirrhosis. We conducted a case-control study in which 164 alcohol-related liver cirrhosis patients and 272 healthy controls were genotyped for the following functional single nucleotide variations (SNVs): *ADH1B* gene, rs1229984, rs1041969, rs6413413, and rs2066702; *ADH1C* gene, rs35385902, rs283413, rs34195308, rs1693482, and rs35719513; *CYP2E1* gene, rs3813867. Furthermore, copy number variations (CNVs) for *ADH1A*, *ADH1B*, *ADH1C,* and *CYP2E1* genes were analyzed. A significant protective association with the risk of developing alcohol-related liver cirrhosis was observed between the mutant alleles of SNVs *ADH1B* rs1229984 (*P_c_* value = 0.037) and *ADH1C* rs283413 (*P_c_* value = 0.037). We identified CNVs in all genes studied, *ADH1A* gene deletions being more common in alcohol-related liver cirrhosis patients than in control subjects, although the association lost statistical significance after multivariate analyses. Our findings support that susceptibility to alcohol-related liver cirrhosis is related to variations in alcohol metabolism genes.

## 1. Introduction

Alcohol consumption is a common habit that varies considerably by location [1]. Recent data of the prevalence of Spanish current drinkers indicate that 55% of females and 78% of males were current drinkers, which is much higher than global data (25% of females and 39% of males) [1]. Excessive alcohol consumption is associated with a wide range of problems relating to physical health, either directly, or through contributions to other health conditions. Consequently, the associated health problems have reached alarming levels, becoming a major public health concern. In 2016, more than 3 million deaths were attributed to alcohol consumption, which represents 1 in 20 deaths worldwide [2]. Excessive alcohol consumption evokes a wide spectrum of hepatic lesions. Steatosis is the earliest and commonest liver disease, which is reversible if the affected individual ceases drinking [3]. However, patients with chronic steatosis are more susceptible to fibrotic liver diseases and 10–20% of heavy drinkers develop the terminal or late stage cirrhosis, which is characterized by excessive liver scarring, vascular alterations, architectural distortion, and eventual liver failure [4].

There is considerable variability in the susceptibility of developing cirrhosis on an individual basis. These determinants reflect the interplay of constitutional and environmental factors. Also, variations in the genes encoding the enzymes playing an active role in ethanol metabolism might be considered as risk factors to develop cirrhosis because impaired ethanol metabolism increases body exposure [5].

Ethanol is predominantly metabolized in the liver, where two different enzymatic systems have been characterized [6]. These are alcohol dehydrogenase (ADH; EC 1.1.1.1) and the microsomal ethanol-oxidizing system (MEOS) [7]. Hepatic ADH consists of five enzyme classes, ADH1 through ADH5. For class I, three subunits α, β, and γ, have been described. The different isoenzymes primarily involved in hepatic ethanol metabolism are homo- and hetero-dimeric molecules, whose subunits are encoded by *ADH1A*, *ADH1B*, and *ADH1C* genes. These isoenzymes catalyze the oxidation and reduction of a wide variety of alcohols into acetaldehydes, with differences in their properties [8]. Subsequently, acetaldehyde is converted by aldehyde dehydrogenases (ALDH; EC 1.2.1.3) to acetate [9]. It is estimated that class I enzymes might contribute to ~70% of the total ethanol oxidizing capacity in the liver [10].

Environmental factors, such as alcohol consumption or concomitant diseases, determine the ARLD progression and the occurrence of alcohol-related liver cirrhosis. Nevertheless, there are interindividual differences in these patients that may not be completely explained by these factors [11]. Thereby, the contribution of genetic factors in ARLDs development has been extensively studied. Substantial interest has been focused on the study of associations between ARLD and variants in genes involved in ethanol metabolism, lipid metabolism, oxidative stress, or immune response [12]. Thus, the variant rs738409 in patatin-like phospholipase domain-containing 3 (*PNPLA3*) has been identified as a risk factor for suffering alcohol-related liver cirrhosis [5,13,14] and non-alcoholic fatty liver disease (NAFLD) [15]. PNPLA3 is a triacylglycerol lipase involved in lipolytic and lipogenic processes [12,16]. Nevertheless, the functional implication of this association has not been completely elucidated. Thus, different studies have demonstrated that the rs738409 *PNPLA3* mutant variant may cause gain [17] or loss of function [18]. In addition, a genome-wide association study also identified two additional gene loci related to lipid metabolism, *MBOAT7* and *TM6SF2*, as risk factor for developing alcohol-related liver cirrhosis [19]. Although these associations are less robust in comparison to that of *PNPLA3,* these variants in genes involved in lipid metabolism, oxidative stress or immune response have been demonstrated to be risk factors both for ARLD and NAFLD [12].

Concerning genes involved in ethanol metabolism, single nucleotide variations (SNVs) are common in *ADH1* genes [20]. These SNVs are associated with changes in enzyme kinetics, which affect production and removal of the toxic metabolite acetaldehyde [21]. These alterations in alcohol pharmacokinetics caused by these SNVs may influence ARLD risk [5]. Thus, it has been shown that the variant rs1229984 in *ADH1B*, which is relatively common among Asians and rarer in Europeans, plays a protective role against alcohol use disorders [5].

The second enzymatic system, MEOS, consists of cytochrome P450, NADPH-cytochrome p450 reductase, and phospholipids. Its activity depends strongly on cytochrome P450 enzymes, predominantly CYP2E1 [8]. CYP2E1 accounts for 20–25% of the alcohol metabolism in vivo [22]. CYP2E1 can be induced by high alcohol levels and following chronic alcohol abuse [23]. The gene variant *CYP2E1*5B,* conformed by two SNVs in close linkage disequilibrium that are placed in the 5’flanking region, modifies the transcriptional activity of this gene [24]. This genetic variant has been associated with ARLD in the Asian population [25,26,27], although this association has not been consistently replicated in Caucasians, probably because of the 10-fold lower frequency of the corresponding SNVs in Caucasian individuals, as compared to individuals with Asian descent [28,29,30,31].

Although most of these above-mentioned studies have been focused on SNVs, there is an increasing interest in gene copy number variations (CNVs) as an additional source of genetic variability. CNVs are structural variations in the DNA sequence consisting of excess or deficiency of sections of DNA sequence [32]. Since these CNVs affect large gene fragments, or the whole gene, their functional consequences are huge. Currently, several studies have analyzed the association between CNVs and alcohol dependence [33,34,35]. Nevertheless, to our best knowledge, no studies analyzing CNVs in genes involved in alcohol biodisposition in alcoholic cirrhosis have been carried out.

Aiming to identify genetic susceptibility factors for the development of alcohol-related liver cirrhosis that could be related to ethanol biodisposition, we analyzed the frequency of *ADH1A, ADH1B, ADH1C*, and *CYP2E1* genetic variants, both SNVs and CNVs, in a well-characterized cohort of Spanish patients.

## 2. Materials and Methods

### 2.1. Human Subject Cohort

A case-control study was designated to investigate the association of functional SNVs and CNVs in the *ADH1A, ADH1B, ADH1C*, and *CYP2E1* genes, with alcohol-related liver cirrhosis. The study included 164 Caucasian Spanish patients with alcohol-related liver cirrhosis and 272 Caucasian Spanish healthy individuals who were recruited at the San Carlos University Hospital (Madrid, Spain) and the University Hospital Infanta Cristina (Badajoz, Spain). Controls were recruited among students and staff. Demographic data, clinical characteristics, and drinking habits were collected for all participants. Table 1 summarizes the characteristics of participants. Concerning healthy individuals, the inclusion criteria were the following: none had personal antecedents of alcoholism or reported familial antecedents of alcoholism, age over 18, absence of consumption of illicit drugs by self-report, and lack of exclusion criteria. As well, the exclusion criteria for healthy individuals were pregnancy, diabetes mellitus, history of gastrointestinal, liver, or renal disease.

For patients, the inclusion criteria were the following: All patients had advanced decompensated ARLD and had consumed at least 100 g of ethanol daily for at least 10 years. Patients were diagnosed based on their liver biopsy. All of them had ultrasonographic patterns compatible with liver cirrhosis and signs of portal hypertension. In addition, the exclusion criteria for patients were the following features: (a) negative results for hepatitis B virus surface antigen and for hepatitis C virus antibodies in serum were prerequisites for their inclusion in the study; (b) absence of other liver disease (autoimmune, metabolic, toxic or drug-induced). All participants were previously informed and gave their informed consent to participate. The protocol for this study was in accordance with the Declaration of Helsinki and its subsequent revisions and was approved by the Ethics Committee of the participating hospitals, University Hospital Infanta Cristina (Badajoz, Spain) and San Carlos University Hospital (Madrid, Spain). Some participants in this study participated in previous studies by our group [36,37,38,39,40,41].

### 2.2. Blood Samples and Genotyping

A 10 mL sample of blood was drawn in an EDTA vacutainer by venopuncture and kept at −80 °C until analyzed. Genomic DNA was extracted from peripheral leukocytes and dissolved in sterile 10 mM Tris HCl, pH 8.0, 1 mM ethylenediaminetetraacetic acid at a final concentration of 400–600 µg/mL.

The SNVs and CNVs tested, selected because of their allele frequencies in the population studied and their expected effect in enzyme activity, are described in Table 2. Genotyping was carried out by using TaqMan assays (Life technologies, Alcobendas, Madrid, Spain), which were designed to detect the previously mentioned SNVs. Detection was carried out in by real-time quantitative polymerase chain reaction in an Applied Biosystems 7500 real-time thermocycler as described by the manufacturer. Full details of the procedure were described previously [42]. All samples were determined in triplicate. Genotypes were assigned by using the gene identification software (7500 software 2.0.3 Applied Biosystems, Foster City, CA, USA) and by analysis of the reference cycle number for each fluorescence curve.

CNVs were analyzed by using the TaqMan copy number assays of the *ADH1A* and *ADH1B, ADH1C* and *CYP2E1* genes, Hs00293646_cn, Hs03076708_cn Hs05919789_cn and Hs00231786_cn, respectively. The procedure was described previously [42]. All assays were designed to hybridize within the open reading frame within the target genes (Life technologies, Alcobendas, Madrid, Spain). The amplification was carried out in an Applied Biosystems 7500 real-time thermocycler as described by the manufacturer, using RNAase P as a copy number reference assay. All reactions were carried out in quadruplicate. Results were analyzed by means of the CopyCaller Software v1.0 (Applied Biosytems, Foster City, CA, USA ) [42]. According to standard procedures in CNV analyses, samples with a single copy of the corresponding gene were named as heterozygous (null/present). Because the probes were designed to detect exonic sequences, even if the rest of the gene would remain intact in these so-called null alleles, the translated protein would not be functional [42].

### 2.3. Statistical Analysis

The descriptive analysis of the different variables analyzed was performed by means of absolute frequencies for categorical variables and mean and standard deviation (SD) for continuous variables using the SPSS 22.0 statistical package (SPSS Inc., Chicago, IL, USA). Kolmogorov–Smirnoff test was used to check normality in the distribution. Then, the Student two-sample *t* test or the Mann–Whitney test were used for continuous variables. The Hardy–Weinberg equilibrium and the linkage disequilibrium analyses were performed with the PLINK v1.07 software (Broad Institute of Harvard & MIT, Cambridge, MA, USA) [43]. The comparison between groups was performed with the Chi-square test and Likelihood ratio test, with an initial crude analysis followed by an adjusted analysis including gender and age as categorical variables. A univariate analysis through logistic enter regression was used to identify independent variables associated with alcohol-related liver cirrhosis. Those variables with a *p* value ≤ 0.05 for the univariate analysis were carried out through to a stepwise logistic multivariate regression. A multivariate analysis through logistic regression using gender and age as covariates was carried out to determine the association of genetic variants, isolated or grouped in haplotypes, and alcohol-related liver cirrhosis status. Adjustments for multiple analyses were performed by using the False Discovery Rate correction. *P_c_* values ≤ 0.05 were considered statistically significant. The association between genetic variants and alcohol-related liver cirrhosis trait was estimated by odds ratio (OR) with a 95% confidence interval (CI) [43].

### 2.4. Availability of Materials and Data

The datasets generated during and/or analyzed during the current study are available from the corresponding author on reasonable request.

## 3. Results

The detailed characteristics of the subjects included in this study are summarized in Table 1. The mean age of alcohol-related liver cirrhosis patients was higher in comparison to healthy subjects, and gender distribution was dissimilar between both groups. This is attributable to the lesser prevalence of alcohol-related liver cirrhosis in women compared to men [44]. However, age is not a relevant factor in allele frequencies, nor is gender as none of the genes studied are located either in X or Y chromosomes. The genotype distribution for all genetic variants tested in both healthy subjects and alcohol-related liver cirrhosis patients was in Hardy–Weinberg’s equilibrium.

The genotype frequencies of the variants examined in the study are shown in Table 3. Considering *ADH1B*, two of the polymorphisms analyzed, rs1041969 Asn57Lys and rs2066702 Arg370Cys, were monomorphic in both cohorts of subjects. In addition, we identified the variant alleles *ADH1B*1* (Arg48 + Arg370) and *ADH1B*2* (His48 + Arg370). Furthermore, heterozygous individuals for the SNV rs6413413 Thr60Ser were identified in both healthy subjects and cirrhosis patients.

Regarding *ADH1C*, three polymorphisms, rs35385902 Arg48His, rs34195308 Pro166Ser, and rs35719513 Pro352Thr, were monomorphic in our study population. Although a low frequency was expected in control individuals, no previous studies analyzed these variants in Caucasian alcohol-related liver cirrhosis patients. The *ADH1C* gene has two major allelic variants, *ADH1C*1* (Arg272 + Ile350) and *ADH1C*2* (Glu272 + Val350). The SNVs responsible for these amino acid substitutions are at a very high linkage disequilibrium [45]. Accordingly, we genotyped the SNV rs1693482 Arg272Glu. The allelic frequency displayed in our study population is in keeping with previous studies in European populations [46].

The results of genotype frequencies observed in individuals stratified by the *CYP2E1* genotype are shown in Table 3.

We found that the frequency of the mutated form of the SNVs *ADH1B* rs1229984 and *ADH1C* rs283413 were significantly lower in alcohol-related liver cirrhosis patients. Concerning *ADH1B* rs1229984, the adjusted *p* value was equal to 0.008. Regarding *ADH1C* rs283413, the adjusted *p* value was equal to = 0.015. For both SNVs, the statistically significant differences remained after FDR adjustment for multiple comparisons (Table 3).

CNV analyses revealed that structural variations for the *ADH1A*, *ADH1B*, *ADH1C*, and *CYP2E1* genes occur in Caucasian individuals. The results obtained are shown in Table 3. Overall findings indicate that *ADH1A* CNVs (in all cases deletions) occur in 1.8% of individuals, being more frequent in alcohol-related liver cirrhosis patients (crude *p* value = 0.013, *P_c_* = 0.944). For *ADH1B*, we identified a single case of a patient with three gene copies, whereas the rest of patients and control individuals carried two copies. For *ADH1C*, we only identified individuals with one or two copies, being less frequent the presence of a single copy in patients with alcohol-related liver cirrhosis, although the differences were not statistically significant, as shown in Table 3. For *CYP2E1,* we identified healthy individuals with one, two, or three copies, whereas all patients with cirrhosis carried two copies of the gene.

Univariate and multivariate regression analysis for alcohol-related liver cirrhosis status based on *ADH1A*, *ADH1B*, *ADH1C*, and *CYP2E1* genetic variants are presented in Table 4. After multivariate logistic analysis, the SNVs *ADH1B* rs1229984 (*p* value = 0.023, β = 0.01) remained associated with the risk of developing alcohol-related liver cirrhosis.

We also investigated the putative effect of haplotypes characterized by the combination of *ADH1B* rs1229984 + rs6413413, and *ADH1C* rs283413 + rs1693482 SNVs. Thirty-one haplotypes were observed in alcohol-related liver cirrhosis patients and healthy individuals (Table 5). Haplotypes that show a significant association and a risk effect for developing alcohol-related liver cirrhosis are those composed of the alleles that also revealed a significant association in single-SNV analyses, thus suggesting that the risk is attributable to the SNVs isolated, rather than to the haplotypes. However, we observed that the combination of the wild type genotype of two SNVs, namely *ADH1B* rs1229984 and *ADH1C* rs283413, was significantly higher in alcohol-related liver cirrhosis individuals when compared with healthy subjects (OR = 6.92, *P_a_* = 0.018, *P_c_ =* 0.078). In addition, the OR associated was higher than that associated with each SNV analyzed separately. This result is keeping in line with the observed protective role of the mutant alleles of both SNVs. However, the association described between this haplotype and the susceptibility for developing alcohol-related liver cirrhosis was found to be non-significant after multiple testing correction.

## 4. Discussions

Alcohol abuse is causing a wide range of hepatic lesions, alcohol-related liver cirrhosis being the most severe and harmful state, which may be lethal [4]. Present knowledge suggests that the susceptibility of developing alcohol-related liver cirrhosis is determined by environmental and genetic factors. Thus, data from twin studies demonstrate the heritability of alcohol dependence and its consequences [5]. The study of variations in genes coding for alcohol metabolizing enzymes could lead to a better understanding of the susceptibility and etiopathogenesis of alcohol-related liver cirrhosis. Thus, we analyzed the association between *ADH1A, ADH1B, ADH1C*, and *CYP2E1* genetic variants and cirrhosis liver disease in a well-characterized cohort of Spanish patients.

Our results suggest that genetic variation in two genes coding for ethanol-metabolizing enzymes, *ADH1B* and *ABH1C*, are related to a lower risk of developing alcohol-related liver cirrhosis. Herein, we described for the first time that *ADH1A* gene deletions were more common in alcohol-related liver cirrhosis patients compared to healthy subjects. Concerning *ADH1B* SNV, rs1041969 and rs2066702 were monomorphic, which is in agreement with the low allele frequency for individuals with European descent (equal to 0.000 and 0.004, respectively, in Southern Europeans according the gnomAD database; https://gnomad.broadinstitute.org/. Accessed on 03 February 2021). Also, the observed allele frequencies for *ADH1B*1 a*nd *ADH1B*2* in healthy controls were in keeping with those reported in Caucasian subjects [30,46,47,48,49]. Moreover, the studies involving the *ADH1B* SNV rs6413413 are scarce. However, the allele frequencies observed in healthy subjects were in concordance with those reported in public databases for Caucasians [50].

Regarding the *ADH1B* rs1229984 (Arg48His) SNV, the *ADH1B*1* (Arg48, Arg370) allele, which encodes for the β_1_ subunit, and the mutated *ADH1B*2* (His48, Arg370) allele that encodes the subunit β_2_, have been described. These two subunits have shown pharmacokinetic differences. The β_2_ subunit shows a 20–40-fold higher V_max_ than the β_1_ subunit [10]. Hence, it could be speculated that the association of the variant *ADH1B*2* allele could be associated with an increased detoxication rate, and hence a lower alcohol exposure. Also, faster ethanol oxidation brings about acetaldehyde accumulation. This fact triggers several unpleasant symptoms including vomiting, headache, and tachycardia. The appearance of these symptoms might act as a disincentive factor to drink alcohol, thereby protecting against ARLDs [5,51]. The *ADH1B* rs1229984 SNV is prevalent in East Asian individuals but is rare in non-Asians [52]. However, the mutated *ADH1B*2* (His48, Arg370) allele has been consistently associated with a protector role against ARLDs in East Asians [51], Africans [53] and Europeans [53]. Thus, our findings are in accordance with previous studies in Asians, where the *ADH1B*2* allele frequency is much higher. Previously, Rodrigo et al. showed that the frequency of the mutated *ADH1B* rs1229984 allele was slightly higher in healthy controls than in alcohol-related liver cirrhosis patients in a Spanish cohort. Nevertheless, this difference was not statistically significant [48]. The lack of association in such study might be due to the small sample size studied. Furthermore, two studies focusing on Spanish men [30] and Spanish women [47] with ARLDs did not find any association of the risk with the SNV rs1229984. However, these two studies analyzed a small and heterogenous alcoholic patients’ cohort, which included cirrhosis, steatosis, or chronic hepatitis, thus calling into question the suitability of these studies to detect significant effects.

Concerning the *ADH1C* gene, the SNVs rs35385902, rs34195308, and rs35719513 frequencies observed in our study agree with the extremely rare occurrence of these SNVs in Caucasians according to public databases [54] and with the frequencies described in the gnomAD database, that were equal to 0.001, 0.000, and 0.001 for the above-mentioned SNVs, respectively. Also, the association studies including the polymorphism rs283413 Gly78X are very sparse. However, the allelic frequency observed in the healthy control cohort was shown in correspondence with the British and Irish population [55]. The occurrence of the mutated *ADH1C* rs283413 allele (Arg78) was statistically significantly higher in healthy controls than in alcohol-related liver cirrhosis patients. It should be stated, however, that the statistical significance of this association is lower than that observed for *ADH1B* rs1229984, the statistical significance after multivariate logistical regression is marginal (Table 4), the SNVs *ADH1C* rs283413 and *ADH1B* rs1229984 are at linkage disequilibrium in all populations (D’ = 0.967), and the linkage is even higher in the Iberian population in Spain (D’ = 1.000) according to the Linkage Disequilibrium Pair Tool (https://ldlink.nci.nih.gov. Accessed on 27 January 2021). Therefore, it cannot be ruled out that the association of the *ADH1C* rs283413 SNV with the risk of developing cirrhosis might actually be due to such a linkage.

Regarding the rest of *ADH1C* SNVs, it has been shown that the *ADH1C*1* variant allele (Arg272 Ile350) encodes the subunit γ_1_ and *ADH1C*2* (Glu272 Val350) the subunit γ_2_. Pharmacokinetic studies demonstrated that subjects carrying *ADH1C*1* can metabolize ethanol at a much faster rate than carriers of *ADH1C*2,* thus resulting in the rapid formation of acetaldehyde [10]. *ADH1C*1* has been associated with the risk of developing ARLD in Asians [56,57], where this allelic variant is more prevalent than in Caucasians [30,46,47]. We did not find any association of this genetic variant with alcohol-related liver cirrhosis patients. Our results are consistent with previous studies in Spaniards and Europeans [30,46,47].

*ADH* gene polymorphisms have been related to the triggering effect of alcohol in migraine attacks [58] and with the risk of developing Parkinson’s disease in women [59], which is related to the effect of alcohol consumption in Parkinson’s disease [60] and with other movement disorders [61].

Regarding *CYP2E1*, we analyzed the variant *CYP2E1**5B rs3813867 (−1295G > C). The genotype frequencies were in correspondence with those described for previous studies in the Spanish population [30] and were similar to the frequencies described in other Caucasian populations [29,31]. This gene variant is located at the 5´regulatory region, and the mutated *CYP2E1**5B allele, rs3813867 (−1295C), is associated with higher transcription and increased enzyme activity [62,63]. The mutant *CYP2E1**5B rs3813867 (−1295C) variant has been associated consistently associated with ARLDs in Asians [26,27,63]. Nevertheless, contradictory results have been reported in Caucasians. Whereas several studies have described this association [29,31,62], other studies did not confirm such association [30,38,47,48]. Our results are in agreement with reports showing no association. Further research is needed to confirm the role of *CYP2E1**5B in Caucasians patients.

CNVs are an important source of variations in the human genome that can affect gene expression by a simple gene-dose effect or can include duplication or deletion of gene regulatory regions [64]. We report for the first time the frequencies for *ADH1A*, *ADH1B*, *ADH1C**,* and *CYP2E1* CNVs in a Spanish cohort of alcohol-related liver cirrhosis patients and in healthy subjects. Our findings show that *ADH1A* CNVs occur at a higher frequency in alcohol-related liver cirrhosis subjects, although the multivariate regression analysis did not reach statistical significance. Further research is needed to explore the clinical relevance of this finding.

We acknowledge the limitation of the patient cohort sample size, which is relatively small considering that the frequency of some of the SNVs analyzed is very low in the population analyzed. In addition, patient and healthy cohorts have demonstrated significant differences in terms of age, gender, or alcohol consumption. To overcome these limitations, comparisons were adjusted for age and gender. However, a limitation still remains because of the lack of heavy drinkers in the control group. Since heavy alcohol consumption is related to the ARLD etiopathogenesis, different alcohol drinking habits between both cohorts may be expected [3]. Besides, this case-control design has been successfully carried out in previous studies to identify genetic risk factors associated to alcohol-related liver cirrhosis [65,66,67]. Concerning the age and gender differences shown between alcohol-related liver cirrhosis patients and controls, all the analyses have been adjusted by these cofounding factors to control possible bias.

In summary, our results show that there is an association between functional SNVs in genes involved in ethanol metabolism and alcohol-related liver cirrhosis. Our findings on *ADH1B* SNVs point to decreased ethanol metabolism as a risk factor of developing alcohol-related liver cirrhosis. On one hand, decreased metabolism leads to higher exposure to alcohol and, on the other hand, decreased metabolism brings about lower production of ethanol metabolites that evoke unpleasant symptoms. With these unpleasant symptoms reduced, higher ethanol consumption or development of chronic alcohol consumption might be expected.

## Figures and Tables

**Table 1 jpm-11-00409-t001:** Characteristics of individuals included in the study.

Variable	Cirrhosis Patients (N = 164)	Controls (N = 272)	*p*-Value
Age (mean ± SD), years	55.83 ± 10.32	23.67 ± 7.38	0.000
Sex (male/female)	145/19	81/191	0.000
Alcohol consumption (mean ± SD), g/day	129.24 ± 62.72	2.14 ± 5.34	0.000

**Table 2 jpm-11-00409-t002:** Selection of SNVs and CNVs genotyped.

Gene	Variant	Consequence
*ADH1B*	CNV	Deletion/Duplication
*ADH1B*	rs1229984	His48Arg
*ADH1B*	rs1041969	Asn57Lys
*ADH1B*	rs6413413	Thr60Ser
*ADH1B*	rs2066702	Arg370Cys
*ADH1B*	CNV	Deletion/Duplication
*ADH1C*	rs35385902	Arg48His
*ADH1C*	rs283413	Gly78X
*ADH1C*	rs34195308	Pro166Ser
*ADH1C*	rs1693482	Arg272Gln
*ADH1C*	rs35719513	Pro352Thr
*ADH1C*	CNV	Deletion/Duplication
*CYP2E1*	rs3813867	Upstream in promoter region
*CYP2E1*	CNV	Deletion/Duplication

**Table 3 jpm-11-00409-t003:** Distribution of genotype frequencies of *ADH1, ADH2, ADH3*, and *CYP2E1* genetic variants.

**Gene**	**Variant** **(SVN id)**	**Cirrhosis Patients:** **Number of DNA Samples, (Frequencies for Non-Mutated; Heterozygous; Homozygous)**	**Controls:** **Number of DNA Samples, (Frequencies for Non-Mutated; Heterozygous; Homozygous)**	**Intergroup Comparison** **OR (95% CI) with Covariates, *P; P_a_; P_c_***
*ADH1B*	rs1229984	151 (0.967, 0.033, 0.000)	250 (0.880, 0.112, 0.008)	0.08 (0.01–0.58), 0.003; 0.008; 0.037
*ADH1B*	rs1041969	128 (1.000, 0.000, 0.000)	259 (1.000, 0.000, 0.000)	--
*ADH1B*	rs6413413	135 (0.993, 0.007, 0.000)	258 (0.988, 0.012, 0.000)	1.66 (0.00–7.15 × 10^5^), 0.685; 0.938; 1.000
*ADH1B*	rs2066702	157 (1.000, 0.000, 0.000)	250 (1.000, 0.000, 0.000)	--
*ADH1C*	rs35385902	160 (1.000, 0.000, 0.000)	229 (1.000, 0.000, 0.000)	--
*ADH1C*	rs283413	152 (0.987, 0.013, 0.000)	240 (0.929, 0.071, 0.000)	0.07 (0.01–0.69), 0.005; 0.015; 0.037
*ADH1C*	rs34195308	149 (1.000, 0.000, 0.000)	253 (1.000, 0.000, 0.000)	--
*ADH1C*	rs1693482	122 (0.377, 0.533, 0.090)	250 (0.400, 0.448, 0.152)	0.92 (0.42–2.03), 0.778; 0.843; 1.000
*ADH1C*	rs35719513	145 (1.000, 0.000, 0.000)	245 (1.000, 0.000, 0.000)	--
*CYP2E1*	rs3813867	154 (0.941, 0.052, 0.007)	250 (0.944, 0.052, 0.004)	0.43 (0.09–2.12), 0.854; 0.301; 0.501
**Gene**	**Variant** **(Structural Variants)**	**Cirrhosis Patients:** **Number of Samples, (Frequencies for One, Two and Three Gene Copies)**	**Controls:** **Number of Samples, (Frequencies for One, Two and Three Gene Copies)**	**Intergroup Comparison** **OR for CNV Carriers (95% CI), *P; P_a_, P_c_***
*ADH1A*	CNV	158 (0.038, 0.962, 0.000)	229 (0.004, 0.996, 0.000)	0.04 (0.00–46.94), 0.013; 0.236; 0.944
*ADH1B*	CNV	156 (0.000, 0.994, 0.006)	231 (0.000, 1.000, 0.000)	22.07 (0.00–1.56 × 10^5^), 0.177; 0.334; 0.668
*ADH1C*	CNV	154 (0.006, 0.994, 0.006)	232 (0.022, 0.978, 0.000)	0.17 (0.00–9.03), 0.336; 0.442; 0.589
*CYP2E1*	CNV	157 (0.000, 1.000, 0.000)	236 (0.008, 0.987, 0.004)	1.13 (0.00–2.84 × 10^10^), 0.633; 0.992; 0.992

*P_a_* values correspond to *p* values adjusted by gender and age. *P_c_* values correspond to *P_a_* values corrected for multiple comparisons by using False Discovery. Rate analyses.

**Table 4 jpm-11-00409-t004:** Univariate and multivariate logistic regression analysis.

		Univariate Logistic Regression			Multivariate Logistic Regression		
Gene	Variant (SVN Id)	*p* Value	β Value (95% CI)	B	*p* Value	β Value (95% CI)	B
*ADH1A*	CNV	0.043	0.11 (0.01–0.93)	−2.20	0.924		-
*ADH1B*	rs1229984	**0.012**	**0.31 (0.12–0.77)**	**−1.18**	**0.023**	**0.01 (0.13–0.72)**	**−2.34**
* ADH1B *	rs6413413	0.695	0.63 (0.06-6.16)	−0.45			
*ADH1B*	CNV	1.0	-	-			
*ADH1C*	rs283413	**0.021**	**0.175 (0.04–0.77)**	**−1.74**	0.090		
* ADH1C *	rs1693482	0.788	1.04 (0.76–1.43)	0.043			
*ADH1C*	CNV	0.377	2.70 (0.30–24.35)	0.992			
*CYP2E1*	rs3813867	1.0	-	-			
*CYP2E1*	CNV	0.641	1.79 (0.15–20.55)	0.581			
Sex	-	**<0.001**	**17.99 (10.44–31.01)**	**2.89**	**<0.001**	**13.38 (3.98–44.95)**	**2.59**
Age	-	**<0.001**	**1.27 (1.21–1.33)**	**0.24**	**<0.001**	**1.25 (1.18–1.33)**	**0.23**

Univariate logistic regression (*p* ≤ 0.2) completed; all statistically significant results then carried through to a stepwise. Logistic multivariate regression analysis (*p* ≤ 0.05). All statistically significant variables from multivariate logistic regression shown in bold type.

**Table 5 jpm-11-00409-t005:** Alcohol dehydrogenase haplotypes in patients with alcoholic liver cirrhosis and healthy controls.

Haplotype Variant (SVN Id)	Genotype	Cirrhosis Patients: Number of Total DNA Samples, (Frequency for Haplotype)	Controls: Number of Total DNA Samples, (Frequency for Haplotype)	Haplotype Association Analysis OR (95% CI), *P; P_a_; P_c_*
rs1229984-rs6413413-rs283413-rs1693482	A-A-G-A	108 (0.349)	228 (0.332)	1.09 (0.48–2.49), 0.464; 0.838; 0.690
rs1229984-rs6413413-rs283413-rs1693482	A-A-T-G	108 (0.006)	228 (0.018)	0.63 (0.03–13.35), 0.179; 0.764; 0.690
rs1229984-rs6413413-rs283413-rs1693482	G-A-G-G	108 (0.013)	228 (0.036)	0.55 (0.05–5.65), 0.081; 0.616; 0.682
rs1229984-rs6413413-rs283413-rs1693482	A-A-G-G	108 (0.631)	228 (0.614)	1.61 (0.71–3.64), 0.393; 0.253; 0.420
rs1229984-rs6413413-rs283413	A-A-T	119 (0.007)	230 (0.026)	0.37 (0.03–4.07), 0.069; 0.417; 0.539
rs1229984-rs6413413-rs283413	G-A-G	119 (0.015)	230 (0.047)	0.27 (0.03–2.27), 0.023; 0.229; 0.420
rs1229984-rs6413413-rs283413	A-A-G	119 (0.978)	230 (0.926)	7.66 (1.56–37.51), 0.002; 0.012; 0.078
rs1229984-rs6413413-rs1693482	G-A-A	111 (0.003)	254 (0.014)	3.08 × 10^−5^ (0.0–0.06), 0.072; 0.007;0.078
rs1229984-rs6413413-rs1693482	A-A-A	111 (0.347)	254 (0.33)	1.03 (0.52–2.05), 0.62; 0.933; 0.723
rs1229984-rs6413413-rs1693482	G-A-G	111 (0.013)	254 (0.042)	0.08 (0.01–1.04), 0.039; 0.054; 0.114
rs1229984-rs6413413-rs1693482	A-A-G	111 (0.636)	254 (0.613)	1.53 (0.68–3.44), 0.517; 0.303; 0.440
rs1229984-rs283413-rs1693482	G-G-A	110 (0.004)	228 (0.019)	4.45 × 10^−4^ (0.00–0.18), 0.076; 0.011; 0.078;
rs1229984-rs283413-rs1693482	A-G-A	110 (0.350)	228 (0.324)	1.13 (0.50–2.57), 0.346; 0.771; 0.690
rs1229984-rs283413-rs1693482	A-T-G	110 (0.005)	228 (0.017)	0.54 (0.02–11.87), 0.132; 0.697; 0.690
rs1229984-rs283413-rs1693482	G-G-G	110 (0.011)	228 (0.033)	0.66 (0.06–7.1), 0.059; 0.730; 0.690
rs1229984-rs283413-rs1693482	A-G-G	110 (0.630)	228 (0.607)	1.64 (0.72–3.75), 0.344; 0.242; 0.420
rs6413413-rs283413-rs1693482	A-G-A	113 (0.349)	228 (0.336)	1.01 (0.48–2.14), 0.664; 0.979; 0.734
rs6413413-rs283413-rs1693482	A-T-G	113 (0.007)	228 (0.025)	0.05 (0.00–0.96), 0.067; 0.047; 0.114
rs6413413-rs283413-rs1693482	A-G-G	113 (0.645)	228 (0.639)	1.43 (0.64–3.20), 0.745; 0.384; 0.525
rs6413413-rs283413	A-T	124 (0.008)	230 (0.037)	0.09 (0.01–0.85), 0.036; 0.035; 0.106
rs6413413-rs283413	A-G	124 (0.992)	230 (0.963)	9.39 (1.15–76.49), 0.041; 0.036; 0.106
rs6413413-rs1693482	A-A	117 (0.348)	254 (0.344)	0.91 (0.41–2.00), 0.924; 0.807; 0.690
rs6413413-rs1693482	A-G	117 (0.652)	254 (0.656)	1.1 (0.50–2.42), 0.917; 0.813; 0.690
rs1229984-rs6413413	G-A	125 (0.016)	258 (0.056)	0.09 (0.01–0.68), 0.021; 0.019; 0.078
rs1229984-rs6413413	A-A	125 (0.984)	258 (0.943)	9.84 (1.43–67.57), 0.027; 0.020; 0.078
rs283413-rs1693482	G-A	115 (0.355)	228 (0.346)	1.08 (0.46–2.55), 0.741; 0.861; 0.690
rs283413-rs1693482	T-G	115 (0.007)	228 (0.024)	0.06 (0.00–1.05), 0.104; 0.054; 0.114
rs283413-rs1693482	G-G	115 (0.638)	228 (0.630)	1.34 (0.60–3.00), 0.729; 0.477; 0.555
rs1229984-rs283413	A-T	138 (0.007)	230 (0.027)	0.39 (0.03–4.40), 0.074; 0.447; 0.547
rs1229984-rs283413	G-G	138 (0.014)	230 (0.054)	0.31 (0.04–2.56), 0.024; 0.276; 0.428
rs1229984-rs283413	A-G	138 (0.978)	230 (0.919)	6.92 (1.39–34.35), 0.001; 0.018; 0.078

*P_a_* values correspond to *p* values adjusted by gender and age. *P_c_* values correspond to *P_a_* values corrected for multiple comparisons by using False Discovery Rate analyses.

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
