# Peer review of "Genetic Variants of Alcohol Metabolizing Enzymes and Alcohol-Related Liver Cirrhosis Risk"

_jpm, 2021, doi:10.3390/jpm11050409_

Round 1

Reviewer 1 Report

The manuscript by Ayuso et al. analyzed the correlation between genetics variants in alcohol metabolizing enzymes and alcohol-related liver diseases. It is interesting to test these associations in a case control study, especially about the copy number variations. However, this study is limited in the study design and data analysis. Please see my specific comments below:

  1. As the authors stated, there is a significant difference between cases and controls in terms of the age, gender, and alcohol consumption. According to the section 2.3, these confounding factors were not adjusted in the statistical analyses either. Therefore, the reported associations between genetic variants and ARLD may suffer from the potential confounding effects.
  2. Although the subjects of the cohort were mainly white Spanish individuals, there may still exist substantial population stratification that would confound the results. Please adjust the population structure in the association analyses.
  3. The details of the univariate and multivariate logistic regression analysis are not clear, please clarify.
  4. Table 5 is not included in the manuscript.

Author Response

Review Report Form (Reviewer 1)

Comments and Suggestions for Authors

The manuscript by Ayuso et al. analyzed the correlation between genetics variants in alcohol metabolizing enzymes and alcohol-related liver diseases. It is interesting to test these associations in a case control study, especially about the copy number variations. However, this study is limited in the study design and data analysis. Please see my specific comments below:

1. As the authors stated, there is a significant difference between cases and controls in terms of the age, gender, and alcohol consumption. According to the section 2.3, these confounding factors were not adjusted in the statistical analyses either. Therefore, the reported associations between genetic variants and ARLD may suffer from the potential confounding effects.

Author response:

We thank the reviewer for this suggestion. To accomplish the significant differences between cases and controls, we have re-analysed the data considering the age and gender as categorical variables and covariates. Thus, the results reported are now adjusted by these two characteristics to avoid potential confounding effects. We have described this statistical analysis in the following lines (L182-L196) and the tables 3, 4 and 5 show the adjusted results. Concerning to the differences in term of alcohol consumption, it is well reported that heavy alcohol consumption is associated with a wide range of hepatic lesions and may be the leading factor of developing alcohol-related liver cirrhosis. We acknowledge that the lack of a healthy control group composed by heavy drinkers is a limitation of our study. However, this case-control design study has been successfully carried out in previous studies to identify genetic variants that may contribute to the alcohol-related liver cirrhosis susceptibility (L416-L425).

2. Although the subjects of the cohort were mainly white Spanish individuals, there may still exist substantial population stratification that would confound the results. Please adjust the population structure in the association analyses.

Author response:

This is an important point that requires clarification and we thank the reviewer. The studied population included only Spanish Caucasian individuals, excluding those participants that reported nationalities and ethnic origin other than Spanish Caucasian. Since a previous population structure analysis developed in Spanish Caucasian individuals did not find major population stratification issues (BMC Genomics. 2010; 11: 326.), we consider that the findings described in our studied population are not affected by population structure bias. We have clarified the population origin in lines L117 and L118.

3. The details of the univariate and multivariate logistic regression analysis are not clear, please clarify.

Author response:

We thank the reviewer for this comment. We have included a statement in the manuscript to detail the regression analysis (L186-192).

4. Table 5 is not included in the manuscript.

Author response:

We are sorry for this and thank the reviewer for this comment. The table 5 has now been included in the main manuscript.

Reviewer 2 Report

The authors aimed to identify the genetic susceptibility for alcohol-related liver cirrhosis by analyzing the frequency of ADH1A, ADH1b, ADH1C, and CYP2E1 genetic variants in a Spanish cohort.

Just small comments regarding the paper:

  • In the methods, the inclusion criteria should be clearly presented for patients and for healthy controls.
  • In the presentation of the results, there are some comments that should be in the Discussions.
  • The authors should explain how/why the controls are so different from the cirrhosis patients (as the age and sex - even that they tried to explain that as being without importance for the analysis). 

Author Response

Review Report Form (Reviewer 2)

Comments and Suggestions for Authors

The authors aimed to identify the genetic susceptibility for alcohol-related liver cirrhosis by analyzing the frequency of ADH1A, ADH1b, ADH1C, and CYP2E1 genetic variants in a Spanish cohort.

Just small comments regarding the paper:

  • In the methods, the inclusion criteria should be clearly presented for patients and for healthy controls.

Author response:

We thank the reviewer for this suggestion. This has been amended and clarified in lines 122-127 for healthy controls and in lines 130-138 for patients.

  • In the presentation of the results, there are some comments that should be in the Discussions.

Author response:

This is a good point and we thank the reviewer for this comment. We have included those comments in the Discussion section (L334-L341, L365-371 and L395-L398)

  • The authors should explain how/why the controls are so different from the cirrhosis patients (as the age and sex - even that they tried to explain that as being without importance for the analysis). 

Author response:

We thank the reviewer for this comment. This is an important point that requires clarification. The alcohol-related liver cirrhosis disease is characterized for being the latest stage of a wide range of hepatic lesions that may be associated with an excessive alcohol consumption. In addition, it has been reported that the percentage of current drinkers is higher in men than in women. However, to avoid that these differences between both cohorts may cause cofounding effects, we have re-analysed the data considering age and gender. Thus, these reported results are adjusted by age and gender. The statistical analysis is described in the following lines (L182-L196) and the tables 3, 4 and 5 show the adjusted results. In addition, we acknowledge the limitation of this study considering the lack of a healthy control group composed by heavy drinkers. Since heavy alcohol consumption is related to the ARLD etiopathogenesis, different alcohol drinking habits between both cohorts are expected, and this issue has been acknowledged as a limitation in this study in the Discussion section (L416-L421, L423-L425). However, this strategy has been previously carried out to evidence the contribution of genetic variants in the risk of suffering alcohol-related liver cirrhosis (L421-L423).

Round 2

Reviewer 1 Report

The authors have addressed all my questions. Thanks!